# Cell Electropermeabilisation Enhancement by Non-Thermal-Plasma-Treated PBS

**DOI:** 10.3390/cancers12010219

**Published:** 2020-01-16

**Authors:** Thai-Hoa Chung, Augusto Stancampiano, Kyriakos Sklias, Kristaq Gazeli, Franck M. André, Sébastien Dozias, Claire Douat, Jean-Michel Pouvesle, João Santos Sousa, Éric Robert, Lluis M. Mir

**Affiliations:** 1Institut Gustave Roussy, Metabolic and Systemic Aspects of Oncogenesis (METSY), Université Paris-Saclay, CNRS, 94805 Villejuif, France; Thai-Hoa.CHUNG@gustaveroussy.fr (T.-H.C.); Franck.ANDRE@cnrs.fr (F.M.A.); 2GREMI, UMR 7344 CNRS/Université d’Orléans, 45067 Orléans, France; augusto.stancampiano@univ-orleans.fr (A.S.); sebastien.dozias@univ-orleans.fr (S.D.); claire.douat@univ-orleans.fr (C.D.); jean-michel.pouvesle@univ-orleans.fr (J.-M.P.); eric.robert@univ-orleans.fr (É.R.); 3Laboratoire de Physique des Gaz et des Plasmas, Université Paris-Saclay, CNRS, 91405 Orsay, France; kyriakos.sklias@u-psud.fr (K.S.); kristaq.gazeli@u-psud.fr (K.G.); joao.santos-sousa@u-psud.fr (J.S.S.)

**Keywords:** cancer, non-thermal atmospheric pressure plasma (NTP), plasma medicine, indirect treatment, plasma-treated phosphate-buffered saline, electroporation, electric pulses, pulsed electric field amplitude, melanoma, long-lived reactive species

## Abstract

The effectiveness of electrochemotherapy (ECT) in local eradication of tumours in human and veterinary medicine has been proven. ECT consists of increasing the uptake of cytotoxic drugs by means of pulsed electric fields (PEFs) that transiently permeabilise the cell membrane. Still, this tumour treatment includes some drawbacks that are linked to the characteristics of the intense electric pulses (EPs) used. Meanwhile, the emerging field of cancer therapies that are based on the application of non-thermal plasmas (NTP) has recently garnered interest because of their potentialities as rich sources of reactive species. In this work, we investigated the potential capabilities of the combined application of indirect NTP treatment and microsecond PEFs (µsPEFs) to outperform in vitro cell electropermeabilisation, the basis of ECT. Thus, phosphate-buffered saline (PBS) was plasma-treated (pPBS) and used afterwards to explore the effects of its combination with µsPEFs. Analysis of two different cell lines (DC-3F Chinese hamster lung fibroblasts and malignant B16-F10 murine melanoma cells), by flow cytometry, revealed that this combination resulted in significant increases of the level of cell membrane electropermeabilisation, even at very low electric field amplitude. The B16-F10 cells were more sensitive to the combined treatment than DC-3F cells. Importantly, the percentage of permeabilised cells reached values similar to those of cells exposed to classical electroporation field amplitude (1100 V/cm) when the cells were treated with pPBS before and after being exposed only to very low PEF amplitude (600 V/cm). Although the level of permeabilisation of the cells that are treated by the pPBS and the PEFs at 600 V/cm is lower than the level reached after the exposure to µsPEFs alone at 1100 V/cm, the combined treatment opens the possibility to reduce the amplitude of the EPs used in ECT, potentially allowing for a novel ECT with reduced side-effects.

## 1. Introduction

Electrochemotherapy (ECT) is a non-thermal, safe, and efficient tumour treatment [1,2,3,4] that is currently used in more than 150 clinics in the European Union and abroad, together with its application in veterinary oncology for treatment of metastases as well as primary tumours [5,6]. ECT is based on the combination of otherwise non- or low-permeant drugs possessing a high intrinsic cytotoxicity (*e.g.*, hydrophilic molecules such as bleomycin or cisplatin) with the local application of a train of eight short and intense monopolar electric pulses (EPs), yet nontoxic [7,8]. The applied EPs create a transient transmembrane potential difference that causes changes in the cell membrane structure and transiently permeabilise its phospholipid bilayer [7]. This biophysical process, which is named reversible electroporation or reversible electropermeabilisation, allows for the penetration of the chemotherapeutic agent inside the cell to generate irreversible DNA damages.

ECT selectively kills the tumour cells at the low doses of the chemotherapeutic agents used, since bleomycin (BLM, of 1415 Da) at low doses is only toxic for the cells dividing in the volume treated by the EPs. Overall, no serious negative effects that are related to the application of ECT on patients have ever been reported. Nevertheless, one of the main drawbacks of ECT application are muscles contraction with discomfort sensations associated with repeated electrical stimulation, mainly linked to the characteristics of the high-amplitude electric pulses used. Indeed, these EPs depolarise the neurons in the treated area and can, therefore, generate action potentials, either in the musculo-excitatory nerves or in the sensory nerves, imposing the use of at least a local anaesthesia during the treatment [9]. Recent clinical studies have reported that the painful sensation that is associated to ECT can last longer for locally advanced and metastatic soft tissue sarcomas [10] and large cutaneous recurrences of breast cancer [11].

The main objective of the present study was, therefore, to determine new conditions for ECT, devoid of these side effects. We were interested in reducing the electric field strength of the classical 100 microseconds pulses used in ECT, without reducing the permeabilisation of the cell membrane. We suggest a combined treatment of µsPEFs with non-thermal plasma (NTP) to outperform ECT since cell electropermeabilisation is characterised by lipids oxidation at the time of the electric pulses delivery [12]. Indeed, NTP can be a source of reactive species favouring the lipid oxidation reactions. Over the last decade, several studies in the plasma medicine field have pointed out the use of plasmas at atmospheric pressure in oncology, as plasmas offer the possibility to achieve cell membrane permeabilisation [13,14] as well as to selectively kill cancer cells without affecting normal cells [15,16,17]. Nowadays, NTP are broadly used not only in preclinical development of anticancer therapies, including malignant melanoma, ovarian, colorectal, liver, lung, hepatoma, breast, and brain cancers [18], but also in clinical studies [19]. The virtue of NTP relies on the abundant production of reactive species that were primarily generated upon plasma-air interactions: reactive oxygen species (ROS), such as superoxide (O_2_^●^¯), hydroxyl radicals (OH●), atomic oxygen (O_2_), singlet delta oxygen (^1^O_2_), ozone (O_3_), and hydrogen peroxide (H_2_O_2_), as well as reactive nitrogen species (RNS), such as nitric oxide (NO), nitrogen dioxide (NO_2_), nitrogen trioxide (NO_3_), nitrous oxide (N_2_O), dinitrogen tetroxide (N_2_O_4_), and also positive ions, such as dinitrogen (N_2_^+^) [20]. Two general NTP strategies are defined for cancer treatment: direct treatment and indirect treatment. The first approach consists of a direct treatment of cancer cells or tumours with the NTP source, where the gaseous plasma species and the plasma-induced electric field have direct actions on the surface of the targets. The second approach implicates an indirect treatment, where the plasma source is used to treat liquids (cell culture media, water, or physiological solutions) and the biological targets are subsequently exposed to the plasma-treated liquids [16,21,22,23,24]. Some of the long-lived ROS and RNS (also known as RONS) that are generated in the plasma-treated liquids (such as hydrogen peroxide, nitrite and nitrate) are known to play a major role in the oxidation of phospholipid bilayers of the cell membrane [25,26]. We speculate that the NTP caused oxidative stress might be an important factor in augmenting the efficacy of the electroporation-based therapies.

Thus, the improvement of cell membrane permeabilisation by the combination of indirect plasma treatment and µsPEFs was explored. Furthermore, we were interested in eventual different responses of two different cell lines to the proposed combined treatment. The study was especially focused on malignant melanoma cells, a very aggressive skin cancer, which is one of the main targets of ECT [8,27,28,29] and plasma medicine [17,30]. For this purpose, we used a novel NTP setup that was able to create multiple plasma jets and, therefore, offer a larger and more homogeneous surface treatment. We also implemented a compensation circuit to change the electrical impedance of in vitro targets to that of a reference model for the human body based on our recent studies on the sample influence on plasma characteristics [31]. This setup was used for the plasma treatment of phosphate-buffered saline (PBS) with Ca^2+^/Mg^2+^ (PBS^+^/_+_). The responses to the combined treatment of the plasma-treated PBS^+^/_+_ (or pPBS) with pulsed electric fields of different strengths were investigated while using adherent DC-3F Chinese hamster lung fibroblasts and adherent malignant B16-F10 murine melanoma cells. Cell membrane permeabilisation was monitored by flow cytometry. The obtained results demonstrate the great potential of the combination of indirect NTP application and µsPEF delivery for cancer treatment.

## 2. Results

We tested the cytotoxic effect of pPBS alone to define favourable initial conditions under which cells would not be excessively harmed by the pPBS prior to the assessment of the effect of the combined treatment (pPBS and EPs). The first assay combining pPBS produced at +7 kV and µsPEF with an amplitude of 1100 V/cm was performed in DC-3F fibroblasts. The results, as shown in Appendix B as a proof of concept, give an initial evaluation of the potential effect of indirect plasma treatment and external electric field pulses. Following these preliminary results, the combined effect of pPBS with PEF was investigated in more detail after the pPBS was further characterised and the experimental conditions were optimised and shown to be reproducible.

### 2.1. Evaluation of the Reactive Species in the Plasma-Treated PBS^+^/_+_

We explored the conditions where the NTP multi-jet source used for the preparation of pPBS would be stable, easy to apply, and its characteristics reproducible over all the treatments of the PBS^+^/_+_, as well as the chemical composition of the resulting pPBS reproducible.

#### 2.1.1. Reactor Electrical Characteristics

3 mL of PBS^+^/_+_ were treated by the NTP that was produced with the multi-jet source. Different treatment times between 1 and 20 min. were studied. The reactor was driven by positive high voltage pulses of either +7 kV or +11 kV peak amplitudes. For both voltage amplitudes, the electrical current of the discharge was monitored continuously during the treatment. The electrical measurements (Figure 1) revealed that the treatment corresponding to +11 kV peak voltage was more stable than that of +7 kV peak voltage over the whole treatment duration. In the frame of this analysis, high-definition videos of the NTP multi-jet operated at +7 kV and at +11 kV (see Appendix A) were also recorded, furthermore supporting the higher stability of the NTP multi-jet at +11 kV. Thus, this value was chosen as the operating voltage to produce plasma-treated PBS^+^/_+_.

#### 2.1.2. Characterisation of the Plasma-Treated PBS^+^/_+_

The concentration of hydrogen peroxide, which is a key player for the peroxidation of lipids [21,32], was first assessed in pPBS prepared while using the NTP setup 1 described in the Materials and Methods. The concentration of H_2_O_2_ increased almost linearly with the treatment time for both voltage amplitudes studied here, i.e., +7 and +11 kV. On top of that, at +11 kV, the concentration of H_2_O_2_ in the pPBS is up to three-fold higher than that in the pPBS that results from a treatment at +7kV (Figure 2a).

After the collection of the proof of concept (Appendix B) and the optimisation of the NTP setup 1, as described in the Materials and Methods (resulting in NTP setup 2), we performed a precise dosimetry of the predominantly stable secondary RONS generated in the pPBS (H_2_O_2_, NO_2_^−^, and NO_3_^−^) as a function of the plasma treatment time (Figure 2b). The reactive radicals accumulation in the pPBS was time-dependant, being the highest when the plasma treatment was the longest (20 min, see Figure 2).

Finally, the pH and the conductivity (σ) of the sham (PBS^+^/_+_ exposed to only the helium flow (no plasma) and compensated for the evaporation with distilled water) and the pPBS after 20 min. of treatment at either +7 kV or +11 kV were also evaluated (Table 1). When compared to the sham, the pPBS displayed a slightly reduced pH and increased conductivity at both +7 kV and +11 kV. The pPBS produced with the plasma source at +11 kV displayed a lower pH and a higher conductivity than that generated at +7 kV.

The plasma treatment time of 20 min. was selected as the condition to use for the preparation of pPBS for all of the following experiments and pPBS was therefore further analysed. The storage temperature and stability overtime of pPBS were investigated (Figure 3). No significant degradation of the previously mentioned RONS was observed over 14 days for a storage temperature of +4 °C.

Altogether, the above results assist in the definition of the optimal NTP multi-jet parameters for the production of the pPBS used in our experiments with cells. Thus, the voltage amplitude of +11kV and the plasma treatment time of 20 min. were used for the chemical activation of the PBS^+^/_+_, which was subsequently stored at +4 °C for up to 14 days.

### 2.2. Investigation of the Effects of the Combined Treatment on DC-3F Chinese Hamster Lung Fibroblasts

Adherent DC-3F Chinese hamster lung fibroblasts were exposed to µsPEF of various amplitudes, 0 V/cm (i.e., no PEF), 600 V/cm, and 1100 V/cm. The µsPEFs were combined with different pre- and/or post-treatments either with sham or with pPBS, as described in Figure 4.

The permeabilisation of the cell membrane was analysed by flow cytometry while using YO-PRO^®^-1 iodide. This approach allows for the determination of the percentage of permeabilised cells (Figure 5a) and the intracellular fluorescence per cell (Figure 5b). In the conditions where no PEF was applied to the cell monolayer (0 V/cm), a slight increase of the YO-PRO^®^-1 uptake was observed in the cells treated while using protocols 2, 4, or 6, as compared to those that were treated with sham. The increase in the intracellular fluorescence intensity of the dye was significant in the case of protocols 2 and 6. When a µsPEF of 1100 V/cm was applied, the percentage of permeabilised cells increased in all of the groups and no significant increase in the percentage of electropermeabilised DC-3F cells was caused by the pPBS treatment, regardless of the treatment protocol (Figure 5a). However, we remarked a significant increase of up to 2.4-fold of the intracellular fluorescence in cells that were treated while using protocol 5, when compared to all of the other protocols and, in particular, to the control (Figure 5b). At 600 V/cm, the results were just the opposite: with respect to the PEF alone, no significant enhancement of the intracellular fluorescence of YO-PRO^®^-1 iodide uptake was caused by the pPBS, regardless of the treatment protocol, but the percentage of electropermeabilised cells displayed a significant increase in the population of cells that were treated under protocols 3, 5, and 6 as compared to the control. Indeed, while only ca. 30 to 35% of cells treated while using sham were permeabilised at 600 V/cm, the population of permeabilised cells treated using protocol 6 was increased by almost 1.7-fold (+69%) and that of the cells treated using protocol 5 was the double (2-fold enhancement). There is no statistically significant difference between protocols 5 and 6. Remarkably, for protocol 5 at 600 V/cm, c.a. 60 to 70% of the treated cells were permeabilised, reaching the same percentage of permeabilised cells as that of the cells in the control that was only exposed to µsPEFs at 1100 V/cm.

### 2.3. Investigations of the Effects of the Combined Treatment on B16-F10 Murine Melanoma Cells

#### 2.3.1. Comparison of the Effect of µsPEF at 600 V/cm versus 1100 V/cm on B16-F10 Cells

We investigated the effect of the combined treatment on B16-F10 melanoma cells while using the same seven protocols of the previous section (Figure 6). Even without any PEF applied, a significant increase of the intracellular fluorescence intensity of the dye was detected for protocols 2, 4, and especially protocol 6. For this protocol 6, even the percentage of permeabilised cells displayed a significant two-fold enhancement as compared to the control. Using PEFs at 1100 V/cm, the percentage of electropermeabilised cells was not statistically different from the control without pPBS, except for protocol 4, which was significantly lower. However, with protocols 5 and 6, a significant increase of up to 2.66-fold of the intracellular fluorescence of YO-PRO^®^-1 iodide was observed as compared to the control. When applying a 600 V/cm PEF, the pre- and post-treatment of cells with pPBS (protocols 5 and 6) induced a significant enhancement of the cell membrane electropermeabilisation, both in the percentage of electropermeabilised cells (up to a 1.8-fold enhancement) and in the fluorescence intensity per cell (up to a two-fold enhancement). There is no statistically significant difference between protocols 5 and 6, both inducing strong cell permeabilisation increase, reaching the same percentage of permeabilised cells as that of the cells that were exposed to 1100 V/cm in the absence of pPBS. We also observed a significant enhancement of the YO-PRO^®^-1 iodide intracellular fluorescence in the cells that were treated at 600 V/cm while using protocol 4, i.e., with only a pre-treatment with pPBS for 20 min.

#### 2.3.2. Comparing the Effect of 500 V/cm versus 1400 V/cm µsPEF on B16-F10 Murine Melanoma Cells

The two previous sections show different behaviours of the two cell lines, particularly in the case of the median intracellular fluorescence while using pPBS and µsPEFs of 600 V/cm amplitude. With the B16-F10 cells being apparently more sensitive to the µsPEF than the DC-3F cells, we decided to investigate the consequences of the application of the seven protocols using µsPEF of only 500 V/cm amplitude. It was also of interest to explore the consequences of using µsPEFs of high field amplitude, as for instance 1400 V/cm, anticipating a larger cell permeability. In this last case, the YO-PRO^®^-1 iodide concentration was reduced to 1µM (instead of 2 µM) to avoid a saturation of the flow cytometer signals.

Once more, when no PEF was applied, simple treatment of the B16-F10 cells with pPBS induced a statistically significant increase, not only in the percentage of permeabilised cells but also in the intracellular fluorescence of YO-PRO^®^-1 iodide. Regarding the percentage of permeabilised cells that were treated with pPBS alone, we observed significant increases when compared to the control: two-fold using protocol 2, 2.2-fold using protocol 4, and 2.8-fold using protocol 6 (Figure 7a). Concerning the intracellular fluorescence of YO-PRO^®^-1 iodide (Figure 7b), we observed a significant 1.7-fold enhancement while using protocol 6, which also showed statistically significant differences with protocols 2 and 4 (there is no statistical difference between protocols 2 and 4 as compared to the control).

At 1400 V/cm, when compared to the control, a significant enhancement of the cell membrane permeabilisation level was observed in cells that were treated while using protocols 1, 3, 5, and 6 in what regards the number of permeabilised cells (up to a 1.5-fold enhancement) (Figure 7a) and while using protocols 3, 4, 5, and 6 in what concerns the intracellular fluorescence of YO-PRO^®^-1 iodide entering the cells (up to a 2.2-fold enhancement) (Figure 7b). No statistically significant difference between protocols 5 and 6 was found at 1400 V/cm. Concerning the PEFs at 500 V/cm, all of the combinations (except protocol 1) resulted in a significant enhancement of the percentage of electropermeabilised cells with respect to the control. The intracellular fluorescence intensity also significantly increased while using protocols 3, 5, and 6 as compared to the control. At this very low µsPEF amplitude and with these cells sensitive to the pPBS alone, a statistically significant difference between protocols 5 and 6 (i.e., between a total contact time of 20 and 30 min. between the cells and the pPBS) was found in the intracellular fluorescence intensity.

Figure 8 illustrates the observed differences in the membrane permeabilisation level of B16-F10 melanoma cells due to the combined treatment. The fluorescence threshold for permeabilised cells was determined from the cells that were treated with sham (the control cells exposed to untreated PBS^+^/_+_) and not exposed to PEF (Figure 8a). We noticed that, even though no PEF was applied (0 V/cm), a slight shift towards higher values of the fluorescence per cell (indicating an increase in membrane permeability) was already present in the population of cells that were treated while using protocol 6 (pre- and post-treatment with pPBS) (Figure 8e when compared to Figure 8a). This corresponds to the significant enhancement of membrane permeabilisation that was observed in Figure 6a and Figure 7a induced by pPBS alone. This shift was larger when low amplitude µsPEFs were applied to the cells using protocol 6 at 500 V/cm (Figure 8f) and 600 V/cm (Figure 8g). At 1100 V/cm (Figure 8h), this shift was much larger and actually the population of positive cells constituted a separate peak. The very high level of fluorescence brought by the PEFs at 1100V/cm indicates a high permeabilisation level, being in agreement with the statistically significant enhancement of the membrane electropermeabilisation levels that were observed in Figure 6 and Figure 7.

## 3. Discussion

In this study, the potential effects of a treatment combining plasma-treated PBS^+^/_+_ (pPBS) and monopolar microsecond pulsed electric fields (µsPEFs) on two different cell lines were investigated. The indirect use of the NTP, by means of the application of pPBS, allowed for us to treat the cells in a very homogenous way. The use of parallel plate electrodes encompassing the whole cell monolayer also brought very homogeneous conditions for the cells’ treatment by the µsPEFs. Finally, great care was put on the cell culture conditions before the application of the pPBS and/or the µsPEFs in order to have homogeneous subconfluently (at ca. 80% density) monolayers. Therefore, all of the conditions were gathered to make possible comparisons between the various protocols that were used in our study.

The combination of the µsPEF and the pPBS aimed at the reduction of the applied electric field strength (without decreasing the cells electropermeabilisation) and/or the enhancement of the electropermeabilisation level of the cell membrane. The range of intensities of the µsPEF that was used in our study has already been demonstrated as non-cytotoxic [33,34]. The work reported in the results section was performed with a setup optimised, as described in the results section and in [31] (NTP setup 2), after establishing the proof of concept (Appendix B) with a new NTP multi-jet setup that was recently developed by our group to homogeneously treat large surfaces (NTP setup 1). The NTP setup 2 produced pPBS with reproducible characteristics due to well-controlled exposure of the PBS^+^/_+_ and allowed for us to examine possible differences in the pPBS effects on the two different cell lines used in this study. Our results indicate that the DC-3F cells are less sensitive than the B16-F10 cells to the pPBS alone. Nevertheless, a statistically significant increase in the intracellular concentration of the YO-PRO^®^-1 could be detected in DC-3F cells, even though the pPBS treatment did not suffice for significantly increasing the percentage of permeabilised cells. A low µsPEF intensity (600 V/cm) combined with a pre- and post-treatment of cells with pPBS favours the permeabilisation of cell membrane and doubles the population of permeabilised cells (as compared to the exposure to only µsPEFs of the same intensity). Furthermore, this combined treatment allows for reaching the same percentage of permeabilised cells as a treatment with µsPEFs alone at 1100 V/cm. However, the level of the intracellular fluorescence is not comparable: this level is much lower at 600 V/cm (even in combination with the pPBS) than at 1100 V/cm. At 1100 V/cm, a further increase, significant (over two-fold enhancement), in the intensity of the fluorescence per cell was observed when the µsPEFs were combined with a pre- and post-treatment of the cells with pPBS. All of these increments reveal an increase of the level of electropermeabilisation by the pPBS.

For the comparison with the DC-3F cells, experiments were also performed with the “cancerous” B16-F10 melanoma cells whose size is larger than that of DC-3F cells. This cell size difference is known to play an important role in the cell membrane electropermeabilisation level [35,36]. Therefore, B16-F10 cells were expected to be more sensitive than the DC-3F cells when exposed to µsPEF of the same amplitude. This was indeed observed at 1100 V/cm, since the median intracellular fluorescence of the B16-F10 cells (about 1.8 × 10^6^ a.u.) was higher than that of the DC-3F cells (0.8 × 10^6^ a.u.), in the absence of pPBS application (Figure 5b and Figure 6b, control conditions). When applying PEFs at low field amplitudes (600 V/cm), this consequence was also noticed (median intracellular fluorescence of 0.55 × 10^6^ a.u. for the B16-F10 cells versus 0.22 × 10^6^ a.u. for the DC-3F cells). These data also point out the increase of the median intracellular fluorescence of both cell lines with higher PEFs intensities. This trend is also observed when the individual flow cytometry diagrams are analysed (Figure 8c versus Figure 8d and Figure 8g versus Figure 8h). The results on the percentage of permeabilised cells also show that B16-F10 cells seem to be more sensitive to both treatments. With a single treatment with pPBS (protocol 6, i.e., 30 min. of total contact time between cells and pPBS) without any PEF application, the percentage of permeabilised B16-F10 cells (Figure 6a and Figure 7a) is significantly higher than the untreated cells (two-fold enhancement), while no statistical difference was found in the case of the DC-3F cells (Figure 5a). At 1100 V/cm for both cells, when the percentage of the µsPEF-permeabilised cells was already very high, this percentage was not increased by the pPBS, regardless of the treatment protocol. Interestingly, when the percentage of µsPEF-permeabilised cells was low (at 600 V/cm for the DC-3F cells and 500 V/cm for the B16-F10 cells—see below), the application of the pPBS resulted in an increase of the percentage of permeabilised cells. Moreover, in a first approximation, the longer the total treatment time with the pPBS, the higher that increase. In the case when the B16-F10 cells were exposed to 1400 V/cm, protocols 5 and 6 again revealed an increase in this percentage, as well as an increase in the intracellular fluorescence intensity, caused by the pPBS application. We can speculate that the increase by pPBS in the cells permeabilisation at 1400 V/cm could be related to the occurrence of longer or irreversible electroporation, which would cause a larger uptake of the YO-PRO^®^-1 iodide by the electroporated cells. To conclude, a pre- and post-treatment of cells with pPBS enhances the membrane electropermeabilisation level for most of the combined treatment protocols with PEFs of 600 to 1400 V/cm amplitude. The oxidative stress that was generated within the phospholipid bilayer of the cells by the radicals brought by the pPBS could mediate this effect. Moreover, a period of 10 min. of pre-treatment of cells with pPBS (protocol 5) is sufficient for starting to generate these effects.

Experiments were repeated at 500 V/cm with the malignant B16-F10 melanoma cells to be under the same extremely low permeabilisation levels by PEF alone as those of the DC-3F cells that were treated at 600 V/cm. Interestingly, even at this low µsPEF amplitude, 10 min. of pre-treatment with pPBS (protocol 3) could already achieve an increase in both the percentage of the µsPEF-permeabilised cells and the intracellular uptake of the dye. A combination of both pre- and post-treatment with pPBS (protocols 5 and 6) strongly induced these effects. Especially, 30 min. of total contact time between the cells and pPBS (protocol 6) resulted in the same percentage of µsPEF-permeabilised cells as the controls µsPEF only treated at 1100 V/cm or 1400 V/cm.

It is worth mentioning that pPBS should affect all of the cells, although slightly, because radicals are present throughout the whole volume of the pPBS in contact with the cells, while µsPEF, particularly at low field amplitudes, should only strongly affect part of the cells (those whose size and geometry allows for locally generating a sufficiently high transmembrane voltage difference). In this respect, we can underline that, regardless of the µsPEF strength applied, it was never observed an entire electropermeabilisation of the cell population for both cell lines studied (i.e., the percentage of electropermeabilised cells did not reach 100% in any of the cell lines that were tested at any treatment condition). This result agrees with most of the in vitro electroporation studies, where the viability of most of the cells is sought [34,35].

Multiple research groups discussed the critical influence of both cell type and cancer type on the cell sensitivity to indirect plasma treatment [16,17,37,38]. On the other hand, it has been documented that different cell types have different responses to oxidative stress [39] or PEFs/electrochemotherapy/electroporation-based therapies [35,36,40,41]. From the “electropermeabilisation” point of view, the most important factor is the size of the cells. Microscope images easily show that the B16-F10 cells are larger than the DC-3F cells. Therefore, it was expected that, in this study, for the B16-F10 cells, lower field amplitudes (500 V/cm) would be necessary to electropermeabilise them to the same extent as the DC-3F cells at 600 V/cm. What we could not anticipate was the increased sensitivity of the B16-F10 cells to the pPBS alone. While the various protocols with pPBS in the absence of PEFs only resulted in a higher intracellular uptake of the YO-PRO^®^-1 for the DC-3F cells, significant differences with respect to the controls were achieved in the case of the B16-F10 cells, concerning both the percentage of permeabilised cells and the intracellular YO-PRO^®^-1 concentration. There is no obvious reason for such a difference.

Importantly, the cell-dependant phenomena that were observed in this study have to be also discussed along with the “plasma-treated PBS” effects. We demonstrate here that the radicals of the pPBS contribute to the enhancement of the cell membrane permeabilisation level in the combined treatment. This observation is in agreement with the study of Vernier and colleagues, who, while using molecular dynamics (MD) simulations and experiments with living cells, demonstrated that electroporating fields target oxidatively damaged areas in the cell membrane [42]. Yusupov and colleagues also used MD simulations to demonstrate that oxidation of the lipids in a phospholipid bilayer lowers the permeation free energy barriers of the ROS, which can further enhance the action of the ROS and also result in a drop of the electric field threshold needed for pore formation (electroporation), with respect to the potential facilitation of the pPBS effects by the PEFs. Their study also highlights that the lipid oxidation by plasma generated ROS synergistically enhances this effect [43].

Regarding the NTP multi-jet that was used in the present study to produce pPBS, there is evidence that the efficacy of plasma-treated liquids depends on the generated RONS concentration, which depends on the operating conditions used for liquid treatment [18,24]. Consolidated data were achieved while using an optimised NTP setup that delivers a stable peak current under very precise geometrical conditions. It is interesting to note that the effects were similar to those that were achieved in the proof of concept (Appendix B) while using a non-optimised setup. In fact, with both setups, the accumulation of H_2_O_2_ in pPBS reached similar levels (ca. 1.4 mM, see Figure 2a,b), which might explain why the increase in the cell permeabilisation level was similar. This fact reinforces the implication of the pPBS radicals in the improvement of the cell membrane permeabilisation by the µsPEF. Moreover, the resulted pPBS can remain stable at +4 °C for later use over a long period (at least 14 days), which facilitates its application and stock production, offering extensive advantages for biomedical purposes as compared to recent studies [44,45,46] It is worth mentioning that there are contradicting observations as to whether the conductivity of an external medium impacts the efficiency of the reversible cell membrane permeabilisation that is caused by PEFs [33,47,48]. However, a study from our group investigating the same DC-3F cell line and µsPEFs, demonstrated that media of lower conductivity induced more efficient reversible permeabilisation [33]. Our results indicate that the observed effects are not linked to the conductivity of the liquids, but rather to the radicals presence, since the pPBS has a higher conductivity than that of the sham (see Table 1) and yet induces more permeabilisation effects on the cell membrane.

As previously mentioned, when cells were treated with pPBS before and after low amplitude µsPEFs, the percentage of permeabilised cells reached values similar to those of cells that were exposed to a classical electroporation field amplitude (1100 V/cm). The combined treatment, thus, opens the possibility to reduce the amplitude of the EP used in ECT, one of the goals of the present study. However, formal proofs have to be brought. Indeed, as demonstrated by the low uptake of the YO-PRO^®^-1 iodide by the cells exposed to the combined treatment at 600 V/cm (DC-3F cells) or 500 V/cm (B16-F10 cells) when compared to the uptake by the cells electropermeabilised by the µsPEFs alone at 1100 V/cm, the intensity of the fluorescent dye in cells that were treated by the combined treatment is low. Nevertheless, because of the large efficacy of BLM once inside the cells (500 molecules are sufficient to kill the dividing cells [49]), the percentage of permeabilised cells is more important than the level of permeabilisation (as quantified by the median intracellular fluorescence). As a matter of fact, in 1988, we already published that the same DC-3F cells, which were exposed to pulses leading to the reversible permeabilisation of 98% of the cells according to Lucifer Yellow uptake (a permeabilisation marker of ca. 450 Da, slightly smaller than the YO-PRO^®^-1 Iodide) also lead to 98% of cell killing in the presence of BLM [7]. In any case, the combined treatment that is explored in this paper is interesting as the µsPEF amplitudes could be greatly reduced during ECT or other electroporation-based therapies (to mitigate their side effects), or the used anticancer drug concentration could be reduced if it is chosen to maintain a high µsPEF amplitude.

## 4. Materials and Methods

Unless specified otherwise, all of the reagents were purchased from Life Technologies, Courtabœuf, France.

### 4.1. Cell Culture

DC-3F Chinese hamster lung fibroblasts [50] and B16-F10 murine melanoma cells [51], all mycoplasma-free, were cultured in Minimum Essential Medium (MEM, 31095-029) and Dulbecco’s Modified Eagle Medium (DMEM, High Glucose, GlutaMAX Supplement, pyruvate, 31966-021), respectively. All of the media were supplemented with 10% foetal bovine serum (FBS, F7524), 100 U·mL^−1^ penicillin and 100 mg·mL^−1^ streptomycin (15140-122). The adherent cells were propagated at 37 °C in a 95% humidity atmosphere containing 5% CO_2_ (HERAcell 240i incubator CO_2_, Thermo Fisher Scientific, Courtaboeuf, FR) and then passaged upon confluency (every two days at a 1:10 dilution or every three days at a 1:30 dilution) while using TrypLE™ Express (12604-013). The cells were routinely checked for mycoplasma contamination via polymerase chain reaction (PCR). Cell viability was assessed while using trypan blue exclusion dye method (Trypan Blue Solution, T8154) with the Countess^™^ II FL Automated Cell Counter (Invitrogen, Thermo Fisher Scientific, Courtaboeuf, FR) and only viable cells were considered.

### 4.2. Plasma-Treated PBS^+^/_+_ Preparation Using NTP Multi-Jet Setups

#### 4.2.1. Specifications of the NTP Multi-Jet Setup

The NTP multi-jet used in this study is based on a Plasma Gun (PG) device, which was described previously [31]. Briefly, the PG is a coaxial dielectric barrier discharge (DBD) reactor that consists of a quartz capillary tube flushed with helium and a microsecond-pulses high voltage (μs-pulses HV) generator powers it. As compared to the classical configuration, this new version presents two reactor zones (Figure 9a). The first zone is located inside the μs-pulsed HV generator, where a high voltage electrode (hollow metallic tube) is placed into a glass tube. At the outlet of the glass tube, a flexible dielectric tube is mounted containing a floating-potential electrode (conductive wire of ca. 1 mm^2^ section) inside it. The dielectric tube goes out from the first reactor zone by connecting it with the second reactor zone. This zone of the plasma device (the applicator) is essentially a second coaxial DBD reactor that is made of a PTFE (Polytetrafluoroethylene) body (shown in white in Figure 9) with the floating electrode being placed in its centre and a grounded ring electrode on the outside. The outlet of the second reactor zone has five micro-orifices (Ø_int_ = 800 µm) producing, thus, five distinct plasma jets and covering a larger liquid area during the treatment (Figure 9b). Four of the orifices are disposed at the corner of a 4.2 mm square with the fifth orifice located at the crossing of the square diagonals. For all of the experiments, helium (99.9999% pure, Air Liquide, FR) at a fixed flow rate of 1 slm (standard litre per minute) was used as the operating gas. The NTP source was powered by high voltage pulses with duration of 4 μs (measured at half height) and a peak value of either +7 or +11 kV at a repetition rate of 2 kHz. The applied voltage was measured while using a high-voltage passive probe (Tektronix P6015A), while the current was determined by measuring the voltage drop on the resistor in the compensation circuit with another high-voltage passive probe (Tektronix TPP1000, Beaverton, OR, USA).

A non-optimised setup 1 was used for cytotoxicity assessment and the proof of concept that is reported in Appendix B (NTP setup 1). It used a stainless steel plate as ground electrode with no compensation circuit and a three-dimensional (3D) printed spacer to fix the gap distance between the plasma output orifices and the surface of the PBS. Based on our concomitant recent work [31], the NTP multi-jet source was optimised (NTP setup 2). It consists of (i) a new and more stable reactor, (ii) a ring shaped wire electrode (stainless-steel wire Ø = 1mm) placed in the bottom of the well (instead of the plane electrode used in the setup 1), which allows for avoiding liquid leakage from the well and reducing the number of discharges that formed between the plasma jets and the grounded electrode, and (iii) a compensation circuit designed to impose the total target impedance to a reference model mimicking the human body impedance [31], which would ease the translation of the results to animal models or human body (Figure 9, note that to allow for better visualisation of the plasma multi-jets, the spacer is not represented in the simplified scheme of Figure 9a and not in use in Figure 9b).

#### 4.2.2. Preparation of the Plasma-Treated PBS^+^/_+_

For the preparation of the liquids used in the indirect plasma treatment with the NTP multi-jet device, Dulbecco’s phosphate-buffered saline with Calcium and Magnesium (DPBS, Ca^++^, Mg^++^, 14040-133), termed PBS^+^/_+_, was chosen, as it is an appropriate buffer for adherent cells. Each well in 24-multiwell plates (Nunclon^®^ Delta Surface, Thermo Fisher Scientific, DK, 142475) was filled with 3 mL of PBS^+^/_+_. The distance between the liquid surface and the plasma multi-jet output orifices was maintained at 4 mm thanks to a customised 3D-printed spacer. PBS^+^/_+_ was exposed to the NTP multi-jet for different times, varying between 1 and 20 min. A digital hygrometer/thermometer (Velleman^®^ Home, BE) was used to monitor the humidity and the room temperature in the working area. Water loss due to evaporation was compensated by distilled water addition in the pPBS at the end of the plasma treatment to maintain the osmolality of the plasma-treated PBS^+^/_+_ (pPBS). Depending on the room temperature, 600 to 700 µL of sterile water were added per well when PBS^+^/_+_ was plasma-treated for 20 min.

### 4.3. Plasma-treated PBS^+^/_+_ Characterisation

The concentration of peroxide (H_2_O_2_) in pPBS was determined while using titanium (IV) oxysulfate (TiOSO_4_). H_2_O_2_ reacts with TiOSO_4_ to produce pertitanic acid, which is yellow [22] and detectable by a spectrophotometer. The absorbance was measured at 407 nm while using a spectrophotometer multi-plate reader (Infinite^®^ M200 PRO Tecan). The quantification of nitrite (NO_2_^−^) and nitrate (NO_3_^−^) in pPBS was performed within an hour after its treatment with the NTP multi-jet using nitrate/nitrite colorimetric assay kits (Cayman Chemical, Interchim, FR), according to the supplier’s instructions and using the same spectrophotometric system was used. The pH and the conductivity of the treated-liquids were also measured while using specific probes for liquid analysis (InLab Micro Pro and Seven Compact Duo, by Mettler-Toledo). Semi-quantitative chemical analyses of peroxide, nitrate, and nitrite concentrations produced in pPBS were also assessed after each pPBS preparation while using Quantofix^®^ test strips (Macherey-Nagel GmbH & Co. KG, Düren, DE) to ensure the quality of each preparation, which also revealed the reproducibility of the pPBS preparation.

For the evaluation of the RONS stability under storage at +4 °C, the initial pPBS volume (3 mL) was divided into four samples of equal volume. The concentration of the reactive species (H_2_O_2_, NO_2_^–^, and NO_3_^–^) in one of the samples was measured in the day of plasma treatment (day 0) and used as a reference (100%). The concentration in the other three samples was measured after 1, 7, and 14 days of storage at +4 °C. The relative concentration of the reactive species after storage at +4 °C is, thus, given in relation to the concentration measured at day 0.

### 4.4. Adherent Cells Electropulsation Setup

We used the system described in [34] for the electropulsation of the adherent cell monolayer. More precisely, as displayed in Figure 10, an in-house built mould of PDMS (polydimethylsiloxane, SYNGARD™ 184 Silicone Elastomer, DE) with an empty 2 cm^2^ rectangle was inserted in a Ø 35 mm Petri dish (Nunclon^®^ Delta Surface, Thermo Fisher Scientific, DK, 353001) to obtain a 2 cm^2^ surface for cell growth. One day before the experiment, 600 µL of cell suspension were added to the defined area of each Petri dish at a density of 2.20 × 10^5^ cells·mL^−1^ (i.e., 1.32 × 10^5^ cells/600 µL) for DC-3F fibroblasts or 1.80 × 10^5^ cells·mL^−1^ (i.e., 1.08 × 10^5^ cells/600 µL) for B16-F10 melanoma cells, in respect to their growing speed and morphology (malignant B16-F10 murine melanoma cells grow much faster than DC-3F Chinese hamster lung fibroblasts). These seeding densities are based on our previous observation and experience, as they appear to be suitable for obtaining a homogenous cell layer at ca. 80% confluency (not entirely dense) after 24 h of cell culture. Different pre-treatment protocols were tested on the day of the experiment. Afterwards, the electric pulses were applied on the cell layer by means of an in-house built electrode configuration consisting of two parallel stainless-steel plates (2 mm thick), fixed in the PDMS mould, and distant of 6 mm (Figure 10). The electrodes and the custom moulding PDMS were designed to ensure the entire exposition of the cell monolayer to EPs. To generate microsecond pulsed electric fields (µsPEFs), the Cliniporator™ (IGEA, Carpi, IT) was used to deliver eight consecutive square-wave electric pulses of 100 µs duration, at a repetition frequency of 1 Hz, and different field strengths (500, 600, 1100, and 1400 V/cm).

### 4.5. Combined Treatment

The combined treatment associated µsPEFs and pPBS to treat cells. Sham (PBS^+^/_+_ not treated with the plasma multi-jet) was used as a control for all of the experiments. Before exposing the cells to the sham or the pPBS, YO-PRO^®^-1 Iodide (629.04 Da) (Life Technologies, Y3603) was added to these media at a final working concentration of 2 µM (unless otherwise specified). In the day of experiment, the medium above the cells was removed and the cell layer was washed twice with PBS^+^/_+_. 500 µL of the YO-PRO^®^-1-containing pre-treatment liquid (sham or pPBS) was added on the cells for different incubation times (10 or 20 min.), and then µsPEFs were delivered (strengths ranging between 500 and 1400 V/cm). Afterwards, the electrodes were removed from the PDMS support and the pre-treatment liquid was then replaced by the YO-PRO^®^-1-containing post-treatment liquid (sham or pPBS) for 10 min.

In the absence of PEF (0 V/cm), the pre-treatment time was fixed at 20 min. In the presence of µsPEFs, the control condition (cells pre- and post-treated using Sham) was carried out with 20 min. of pre-treatment.

### 4.6. Evaluation of Cell Membrane Electropermeabilisation Induced by the Combined Treatment

The eventual permeabilisation of the cells that were treated by the combined treatment was investigated by fluorescent nucleic acids stain YO-PRO^®^-1 Iodide uptake. YO-PRO^®^-1 is a non-permeant dye that is frequently used as an indicator for permeabilisation. Indeed, when the cell membrane is permeabilised, YO-PRO^®^-1 can enter the cell and intercalates with nucleic acids, which results in a strong green fluorescence signal that can be detected by flow cytometry.

After the post treatment, the treated liquid above the cell monolayer was removed and the cells were harvested while using TrypLE™ Express Enzyme dissociation (400 µL per Petri dish). The cell suspension was then analysed by flow cytometry (C6 flow cytometer, BD Accuri, San Jose, California, US). 10 000 events were recorded, the YO-PRO^®^-1 uptake (cell permeabilisation) was evaluated while using green fluorescence channel (excitation 488 nm, emission 530/30 nm). More precisely, the percentage of fluorescent/permeabilised cells as well as the mean and median fluorescence intensity per cell (level of permeabilisation of each cell) were assessed. The cell membrane permeabilisation threshold was fixed for all of the samples based on that of cells treated with Sham and not exposed to PEF.

### 4.7. Statistical Analysis

The experiments were performed at least three times independently, i.e., at least over three different days. In addition, each parameter set was performed in triplicate, which resulted in a total of at least nine replicates for each parameter set. The outliers were identified and removed while using the Grubb’s method (Alpha = 0.05). To study the significance of differences, one-way ANOVA, followed by Bonferroni’s multiple comparison tests, were performed while using Prism (GraphPad Software, La Jolla, CA, US). Statistical significance levels were associated with *p*-values of <0.05 (*), <0.01 (**), <0.001 (***) and <0.0001 (****).

## 5. Conclusions

The present study demonstrates that the application of plasma-treated PBS^+^/_+_ that is produced by the newly-developed NTP multi-jet source could effectively enhance the µsPEFs induced cell membrane permeabilisation, both in terms of the percentage of permeabilised cells and the intracellular content of a permeabilisation marker. These effects occur, even at very low µsPEF amplitudes, and the results are cell-dependant. The malignant B16-F10 murine melanoma cells are more sensitive to the effect of this combined treatment than the DC-3F Chinese hamster lung fibroblasts.

These very promising results underline the great potential of a combined ECT and indirect NTP treatment for anticancer therapies. Investigation and in-depth physical, chemical, and biological understanding of those effects will be fundamental for in vivo studies and clinical trials, which might open up new ways for the implementation of guided cancer therapies while using indirect NTP treatment and ECT in the future.

## Figures and Tables

**Figure 1 cancers-12-00219-f001:**
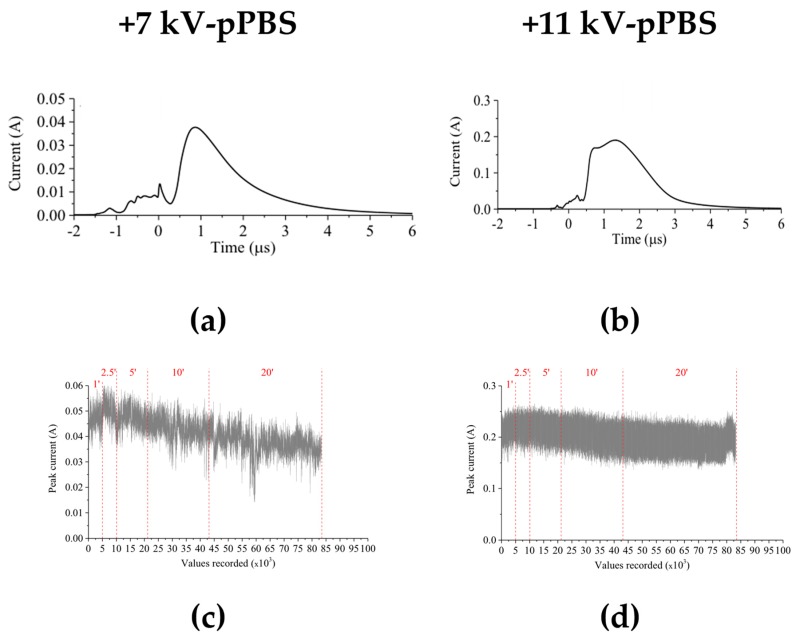
Assessment of the electrical characteristics of the non-thermal plasma (NTP) multi-jet source during 20 min. treatment of PBS^+^/_+_ at +7 kV (**a**,**c**) and + 11 kV (**b**,**d**) pulses peak values. Current signals, averaged from 128 single recordings (**a**,**b**) and the evolution of the maximum current (**c**,**d**) were measured during the treatment process.

**Figure 2 cancers-12-00219-f002:**
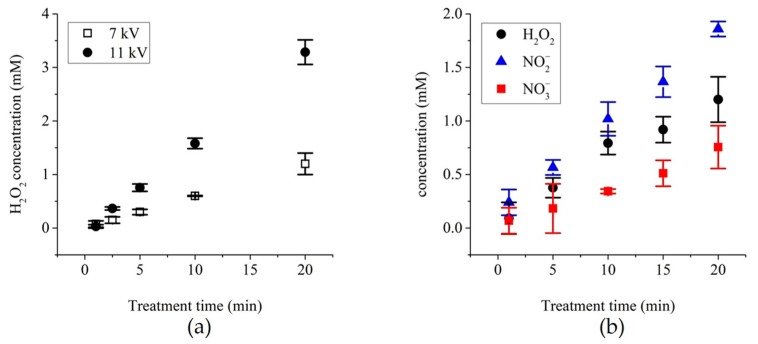
Chemical analysis of the plasma-treated PBS (pPBS) prepared with the NTP multi-jet source using (**a**) NTP setup 1 and (**b**) NTP setup 2. (**a**) The evolution of the concentration of H_2_O_2_ in +7 kV- and +11 kV-pPBS as a function of the treatment time with the plasma, (**b**) Dosimetry of H_2_O_2_, NO_2_^−^, and NO_3_^−^ levels in PBS^+^/_+_ treated during 20 min. at +11 kV. Data are presented as mean values ± SD of independent duplicates ((**b**): 1, 5, and 15 min.) or triplicates (all other points).

**Figure 3 cancers-12-00219-f003:**
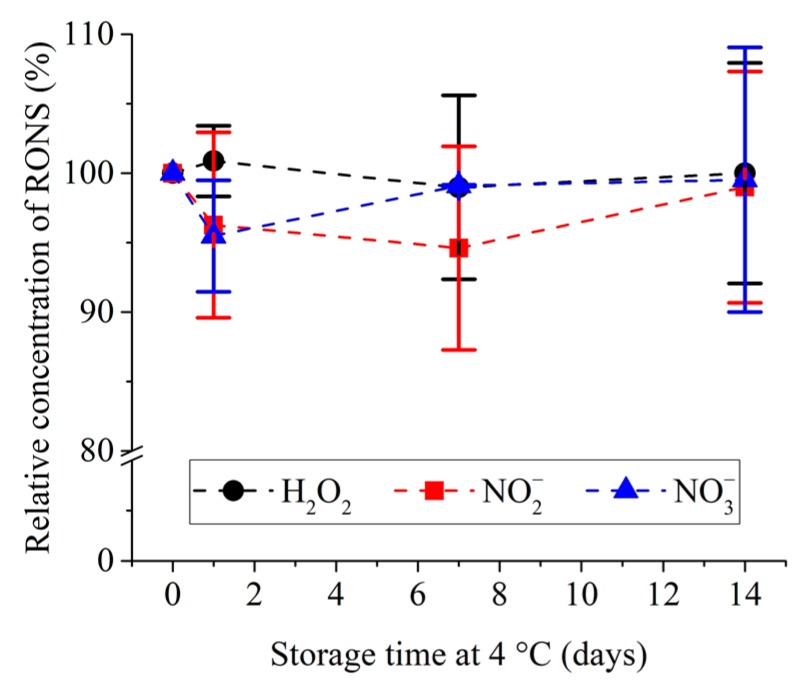
Relative concentration of H_2_O_2_, NO_2_^–^ and NO_3_^–^ in pPBS treated with NTP setup 2 at +11 kV during 20 min. and stored afterwards at +4 °C for up to 14 days. The reference (100%) is the concentration measured after plasma treatment at day 0. Data are presented as mean values ± SD of independent quadruplicates.

**Figure 4 cancers-12-00219-f004:**
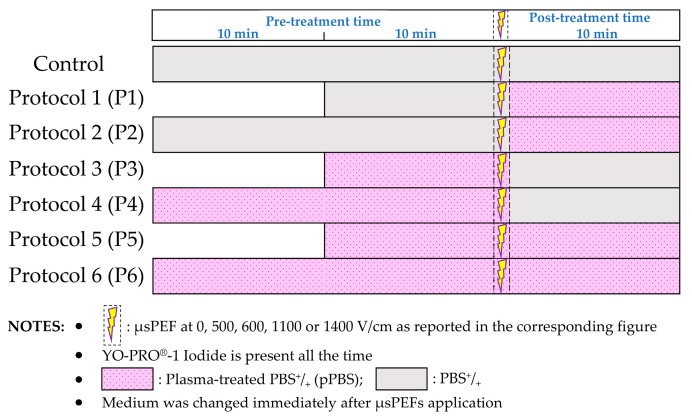
Schematic illustration of the protocols applied in the combined treatments, with µsPEF at 0, 500, 600, 1100 or 1400 V/cm.

**Figure 5 cancers-12-00219-f005:**
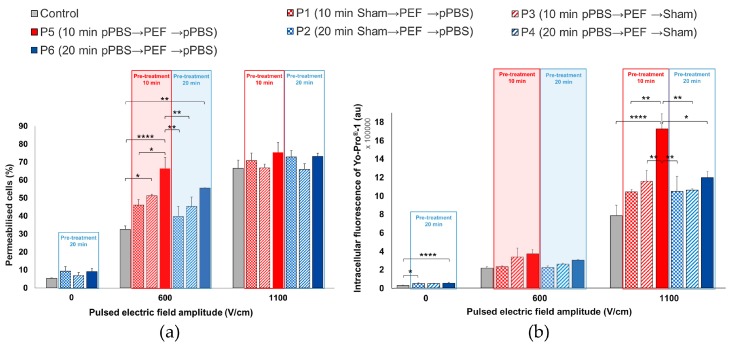
Effects of the combined treatment on DC-3F fibroblasts at 0, 600, and 1100 V/cm. (**a**) Percentage of electropermeabilised cells and (**b**) intracellular fluorescence of YO-PRO^®^-1 iodide entering the cells as a function of the 7 combined protocols applied. The data are presented as mean (for **a**) and median (for **b**) values ± SD of independent triplicates. Statistical differences were analysed while using One-way ANOVA followed by Bonferroni’s multiple comparison test. * *p* < 0.05, ** *p* < 0.01, and **** *p* < 0.0001 significant differences.

**Figure 6 cancers-12-00219-f006:**
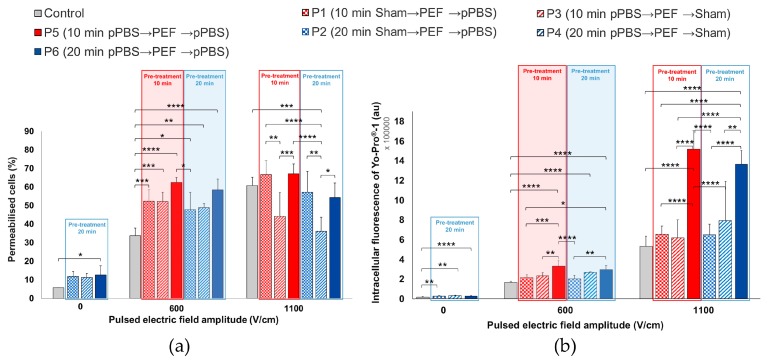
Effects of the combined treatment on malignant B16-F10 melanoma cells using µsPEF at 0, 600, and 1100 V/cm. (**a**) Percentage of electropermeabilised cells and (**b**) intracellular fluorescence of YO-PRO^®^-1 iodide entering the cells as a function of the seven combined protocols applied. Data are presented as mean (for **a**) and median (for **b**) values ± SD of independent triplicates. Statistical differences were analysed while using One-way ANOVA followed by Bonferroni’s multiple comparison test. * *p* < 0.05, ** *p* < 0.01, *** *p* < 0.001, and **** *p* < 0.0001 significant differences.

**Figure 7 cancers-12-00219-f007:**
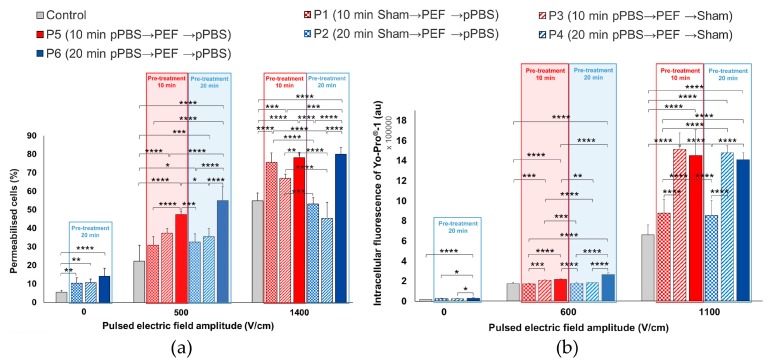
Effects of the combined treatment on adherent malignant B16-F10 melanoma cells using µsPEF at 0, 500, and 1400 V/cm. (**a**) Percentage of electropermeabilised cells and (**b**) intracellular fluorescence of YO-PRO^®^-1 iodide entering the cells as a function of the seven combined protocols applied. Data are presented as mean (for **a**) and median (for **b**) values ± SD of independent triplicates. Statistical differences were analysed using One-way ANOVA followed by Bonferroni’s multiple comparison test. * *p* < 0.05, ** *p* < 0.01, ***, *p* < 0.001, and **** *p* < 0.0001 significant differences.

**Figure 8 cancers-12-00219-f008:**
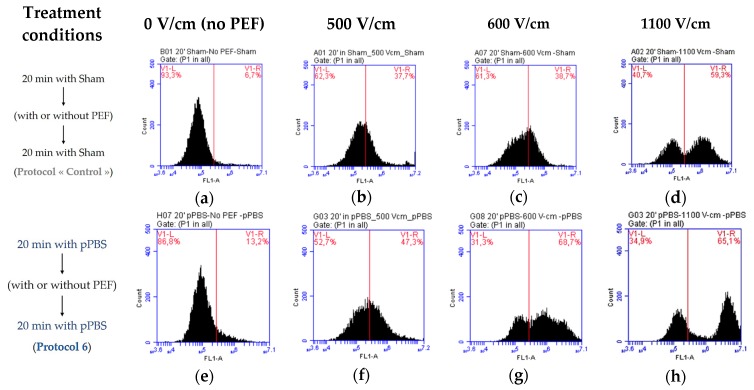
Flow cytometry analysis of a few interesting conditions of the combined treatment in adherent B16-F10 murine melanoma cells to compare effects of protocol 6 (**e**–**h**) versus the control (**a**–**d**) with µsPEFs of 0 (**a**,**e**), 500 (**b**,**f**), 600 (**c**,**g**), and 1100 (**d**,**h**) V/cm. The peak of count as a function of the green fluorescence (FL1-A) shifts towards higher values of the fluorescence per cell, indicating the enhancement of the permeabilisation of the cell membrane.

**Figure 9 cancers-12-00219-f009:**
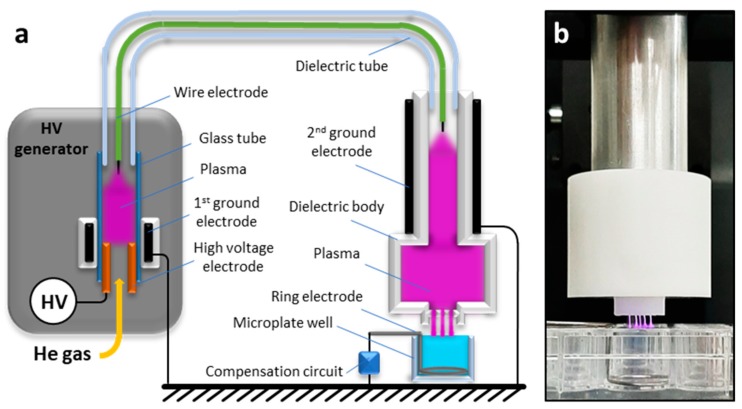
Schematic representation (**a**) and photo (**b**) of the NTP device with a multi-jet nozzle.

**Figure 10 cancers-12-00219-f010:**
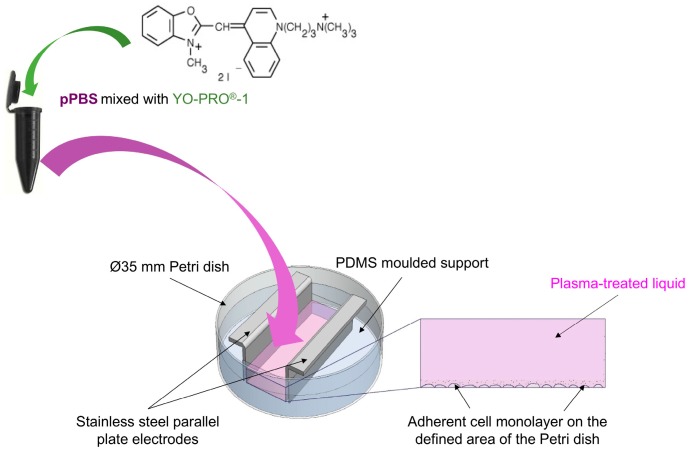
Schematic illustration of the electropulsation setup used in the combined treatment.

**Table 1 cancers-12-00219-t001:** pH and conductivity (σ) of the control (sham) and the pPBS at +7 kV and +11 kV for a PBS^+^/_+_ treatment time of 20 min. Data are presented as mean values ± SD of independent quadruplicates.

	Control (sham)		+7 kV pPBS		+11 kV pPBS	
	Mean	SD	Mean	SD	Mean	SD
**pH**	7.18	0.05	6.68	0.02	6.24	0.10
**σ** (S/m)	10.7	0.08	11.27	0.21	13.08	0.27

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
