# Peer review of "Cell Electropermeabilisation Enhancement by Non-Thermal-Plasma-Treated PBS"

_cancers, 2020, doi:10.3390/cancers12010219_

Round 1
Reviewer 1 Report
Thai-Hoa Chung, et al. investigated the enhanced cell electropermeabilization by the non-thermal-plasma-treated liquid. The authors investigated the potential capabilities of the combined application of indirect NTP treatment and the microsecond PEFs to outperform in vitro cell electropermeabilization, the basis of ECT. The plasma-treated PBS was used afterward to explore the combined effects of indirect NTP and PEFs. Two cell lines, fibroblasts and murine melanoma cells B16F10 were investigated in this study. They found a significant increase in the cell membrane electropermeabilization, even at a very low electric field amplitude. Cancer cells are more sensitive to combined treatment than normal cells.
I have the following recommendation for a major revision.
The ‘liquid’ is not a precise description here. The authors actually just used PBS. Please use ‘PBS’ or ‘phosphate-buffered saline’ to replace ‘liquid’. Please change the structure of this paper. ‘Materials and Methods” should be moved and should be followed by ‘Results and Discussions’. The plasma-activated PBS has been used in previous studies about cancer treatment. Please introduce them in the introduction part:
Van Boxem, Wilma, et al. "Anti-cancer capacity of plasma-treated PBS: effect of chemical composition on cancer cell cytotoxicity." Scientific reports 7.1 (2017): 16478.
Yan, Dayun, et al. "The specific vulnerabilities of cancer cells to the cold atmospheric plasma-stimulated solutions." Scientific reports 7.1 (2017): 4479.
Girard, Pierre-Marie, et al. "Synergistic effect of H 2 O 2 and NO 2 in cell death induced by cold atmospheric He plasma." Scientific reports 6 (2016): 29098.
To facilitate the understanding of the audiences, I recommend the authors use a schematic illustration in the Methods and Materials to explain the protocols (control, p1, p2, p3, and p4) more clearly.
To facilitate the understanding of the audiences, the bars of control, p1-p6 should be drawn in different colors, rather than in the current form (in all figures involving these bars).
According to the statement of the authors, the ‘pPBS’ may mix with YO-PRO-1 or not. But, this information has been revealed in the explanation of Fig. 4, 5, and 6. I recommend the authors clearly marked whether pPBS with or without YO-PRO-1 in these three figures. For example, P5 (10 min pPBS – PEF – pPBS). Are the two pPBS the same? All with or without YO-PRO-1? If they all contain YO-PRO-1, why the authors used the pPBS to affect the cells again after the PFE treatment?
Basically, this paper just focused on the penetration of YO-PRO-1 in the cancer cells or normal cells after the synergistic treatment of pPBS with PEF. For cancer treatment, we more care about the anti-cancer effect, rather than just the penetration of fluorescent material. I recommend the authors also investigate whether the cell viability has been affected by PEF, because pPBS is a potential drug-like method for cancer treatment.
In addition, if possible, I recommend the authors demonstrate that PEF can indeed increase the penetration of (at least) an anti-cancer drug in B16F10 cell line using the same strategy shown in this study. The drug's anti-cancer effect should be least increased by using PEF.
Author Response
Dear reviewer,
Many thanks for your prompt review and suggestions.
As suggested, “liquid” has been replaced by “PBS” for precision. We respected the template (the structure of the paper) of Cancers journal, therefore the section Materials and Methods was placed after that of Results and Discussion, as requested. As suggested, the 3 references regarding the investigations of “plasma-treated PBS” have been introduced in the Introduction section (References at line 83). We thank you to show us that our text was not explicit enough. Changes have been introduced in section 4.5. to describe more precisely the combined treatments. We thank you for your suggestion. Table 2 has now been replaced by Figure 4 which is more clear than the Table. We have used two very different colours to clearly distinguish between the 10 and 20 minutes incubation before the µsPEF applications. Then, with each group, the filling of the histograms is different for each treatment condition. We found this combination the most efficient to allow clear distinction between the controls and P1 to P6. Not that one of the co-authors is colour-blinded (daltonism) and this person is totally unable to distinguish 7 colours, while the combination of 2 colours and 4 fillings does cause no trouble. Therefore we have not modified the figures. The range of intensity of the µsPEF used in our study has already been demonstrated as non-cytotoxic for the cells [Silve A et al. 2016; Ghorbel A et al. 2019]. We thank you for your remark. This statement has been introduced to the revised version of the manuscript (lines 293-294). Even though our study used YO-PRO®-1 Iodide as a cell membrane permeabilisation marker, we justified why the enhancement of the cell membrane electropermeabilisation level will be also effective to increase the penetration of the anticancer drugs used in ECT, even at a lower dose than the current dose used in ECT, thanks to the proposed combined treatment of µsPEF and pPBS (revised manuscript, line 410 to 416). Indeed, many studies of electroporation-based therapies have already shown the cellular sensitivity of anticancer drugs such as bleomycin (BLM, mentioned in the Introduction of the manuscript). We demonstrated years ago that 500 molecules of BLM entering the cell at a given moment (during or just after the electric pulses application) are sufficient to kill diving cells [Poddevin B et al. 1991]. On the contrary, to overcome the cells auto-fluorescence, a large number of fluorescent molecules (g. YO-PRO®-1 Iodide) have to be internalised to get a positive signal in the permeabilised cells.We thank you again for your time and interest in our work.
Best regards,
Reviewer 2 Report
Authors showed a very interesting and somewhat promising technology in treating cancers by combining pulsed electric field and cold plasma to cancer cells. And, argues this work provides evidence in lowering the PEF intensity for the treatment. The description of the procedures and the results are very well organized and well presented.
Major Concerns,
Line 51, The authors stated that ECT selectively kills the tumor cells since low dose of bleomycin is only toxic to the dividing cells, that does not differentiate dividing tumor cells and dividing healthy cells.
For the results presented in Fig 4-6, clearly, 110 V/cm shows much more intracellular fluorescence in different cells, and the authors concluded that this work could provide evidence on lowering the PEF. Please justify.
Comparing Fig 7h and 7d, with the greatest effect from 110 V/cm there is only around 50% of the cells increased the intracellular fluorescence content, does it represent the treatment efficacy of this method?
Author Response
Dear reviewer,
Many thanks for your review.
Line 51: Most healthy cells in the body are not actively dividing. Healthy cells divide under appropriate external signals (growth factors) and favourable environmental conditions, which is not the case of cancer cells who constantly divide. Moreover, without exhibiting contact inhibition, cancer cells continue to divide without receiving the “all clear” signal, as opposed to (dividing) healthy cells. In the treated volume, this already involves an important distinction between healthy and cancer cells and a clear difference in their survival (all cancer cells disappear in 8 to 10 weeks). Moreover, thanks to the locally applied electrodes at the tumour site, electrochemotherapy selectively targets the treated volume which mainly contains actively dividing cancer cells… There are two different aspects which both have been discussed in the manuscript (in particular in lines 398 to 409): The percentage of permeabilised cells and the level of permeabilisation (reflected by the median value of the cellular fluorescence). Bleomycin (BLM) is extremely effective as the penetration of 500 molecules is sufficient to kill the dividing cells. Thus, what is important is the proportion of permeabilised cells and not the level of their permeabilisation. Therefore, for BLM, it is clear that our results indicate that high field intensity pulses can be replaced by lower field amplitude pulses if they are combined to the pPBS treatment. The Figures 7d and 7h refer to results obtained in Figure 5 of the initial submission (becoming Figure 8d and 8h - referring to results obtained in Figure 6 - of the revised one), at 1100 V/cm (60% of the cell population electropermeabilised, that is under conditions fully (or almost fully) preserving the viability of the cell population exposed to the µsPEF). We choose to present the results at 1100 V/cm and not at 1400 V/cm (the “optimal” field in vitro) because, overall, our study aimed to show the complementary of the treatments (that is the increase of PEFs efficacy by the combined treatment with pPBS). When treating tumours, the appropriate field strength is applied, according to the Standard Operating Procedures that we published in 2006 (Mir LM et al.), updated in 2018 (Gehl J et al.). On the average, 78% of the treated tumour nodules disappear after 1 single session of ECT and 10 to 15% more are in partial regression 6 months later.We thank you again for your time and interest in our work.
Best regards,

Reviewer 3 Report
The manuscript entitled “Cells Electropermeabilisation Enhancement by Non-Thermal-Plasma-Treated Liquid” submitted by Chung et al. The concept of the paper is fine; however, the research data to support results is weak in the current form.
Comments:
Author should present strong evidence for their presented concept. Author must provide experimental results related to microscopic images of cells (to show permeabilisation, immuno-cytochemistry ICC images or fluorescence images) and membrane surface markers affected during combination treatment. Minor Comment: Author may improve font size inside figure 4, 5,and 6. Author must improve quality of figures using professional software’s.Author Response
Dear reviewer,
Many thanks for your review and suggestions.
Membrane thickness is 5 nm. No change in the membrane structure is detectable in microscopic images of the electroporated cells (the “holes” or “electropores” have NEVER been seen). It is well established in the PEF research community that the “holes” shown in SEM after cells electroporation and cryofracture (Chang 1992) were artefacts. Similarly, a cell membrane can be electropermeabilised and shows no significant change in surface markers distribution. Font size and histogram size have been homogenised in the revised manuscript, as suggested.We thank you again for your time and interest in our work.
Best regards,

Round 2
Reviewer 1 Report
Thanks for the modifications from the authors.
This is a problem with the reference citation.
The authors stated that "Multiple research groups discussed the critical influence of both cell type and cancer type on the 359 cell sensitivity to indirect plasma treatment [16,17,37–39]." The reference 38 has nothing to do with the indirect plasma treatment. It is about direct treatment. I recommend the authors delete it.
Author Response
Dear reviewer,
Thank you very much for your very careful reading of the revised version of the manuscript.
Reference 38 has been deleted. The numbers of the other references (above 38) have been modified accordingly.
Thank you for having helped us to improve the manuscript.
Sincerely yours
The authors
Reviewer 2 Report
The authors have done a great job revising the manuscript. The study will be an important addition to the field.
Author Response
Dear reviewer,
Thank you very much.
Sincerely yours
The authors
Reviewer 3 Report
I recommend to accept this manuscript as author already revised it carefully.
Author Response

(The authors gave the same response as above.)
